# A Computational Framework for Comprehensive Genomic Profiling in Solid Cancers: The Analytical Performance of a High-Throughput Assay for Small and Copy Number Variants

**DOI:** 10.3390/cancers14246152

**Published:** 2022-12-13

**Authors:** Luciano Giacò, Fernando Palluzzi, Davide Guido, Camilla Nero, Flavia Giacomini, Simona Duranti, Emilio Bria, Giampaolo Tortora, Tonia Cenci, Maurizio Martini, Elisa De Paolis, Maria Elisabetta Onori, Maria De Bonis, Nicola Normanno, Giovanni Scambia, Angelo Minucci

**Affiliations:** 1Bioinformatics Core Facility, Gemelli Science and Technology Park (G-STeP), Fondazione Policlinico Universitario A. Gemelli IRCCS, 00168 Roma, Italy; 2UOC Gynecologic Oncology, Dipartimento per le Scienze della Salute della Donna, del Bambino e di Sanità Pubblica, Fondazione Policlinico Universitario A. Gemelli IRCCS, 00168 Roma, Italy; 3Faculty of Medicine and Surgery, Università Cattolica del Sacro Cuore, 00168 Roma, Italy; 4Direzione Scientifica, Fondazione Policlinico Universitario A. Gemelli IRCCS, 00168 Roma, Italy; 5Comprehensive Cancer Center, Fondazione Policlinico Universitario Agostino Gemelli IRCCS, 00168 Roma, Italy; 6UOC Anatomia Patologica, Dipartimento per le Scienze della Salute della Donna, del Bambino e di Sanità Pubblica, Fondazione Policlinico Universitario A. Gemelli IRCCS, 00168 Roma, Italy; 7Genomics Core Facility, Gemelli Science and Technology Park (G-STeP), Fondazione Policlinico Universitario A. Gemelli IRCCS, 00168 Roma, Italy; 8Cell Biology and Biotherapy Unit, Istituto Nazionale Tumori “Fondazione G. Pascale”-IRCCS, 80131 Napoli, Italy

**Keywords:** next-generation sequencing, bioinformatics analysis, panel validation, coverage analysis, oncology

## Abstract

**Simple Summary:**

Comprehensive genomic profiling (CGP) is key to characterizing solid tumors at the molecular level and enabling personalized therapy. To this end, Fondazione Policlinico Universitario Agostino Gemelli IRCCS launched a CGP program enrolling cancer patients who were screened for nine different solid tumors (breast, colon, GIST, lung, melanoma, ovary, pancreas, prostate and thyroid). In this context, we evaluated the performance of the Illumina^®^ TSO500 high-throughput assay.

**Abstract:**

In January 2022, our institution launched a comprehensive cancer genome profiling program on 10 cancer types using a non-IVD solution named the TruSight Oncology 500 Assay provided by Illumina^®^. The assay analyzes both DNA and RNA, identifying Single-Nucleotide Variants (SNV)s and Insertion–Deletion (InDel) in 523 genes, as well as known and unknown fusions and splicing variants in 55 genes and Copy Number Alterations (CNVs), Mutational Tumor Burden (MTB) and Microsatellite Instability (MSI). According to the current European IVD Directive 98/79/EC, an internal validation was performed before running the test. A dedicated open-source bioinformatics pipeline was developed for data postprocessing, panel assessment and embedding in high-performance computing framework using the container technology to ensure scalability and reproducibility. Our protocols, applied to 71 DNA and 64 RNA samples, showed full agreement between the TruSight Oncology 500 assay and standard approaches, with only minor limitations, allowing to routinely perform our protocol in patient screening.

## 1. Introduction

The implementation of cancer molecular characterization in clinical practice has improved prognostic redefinitions, extending their eligibility to a continuously increasing number of targeted treatments. Broad molecular profiling technologies better than organ-based approaches are believed to serve such dynamic purposes. Next-generation sequencing (NGS) approaches have progressively allowed the spread of such determinations, facilitating the execution of comprehensive genomic profiling (CGP) assays on patients’ tumor samples [1,2]. Specifically, CGP identifies molecular alterations that sometimes can be targeted by the available treatments [1,2].

The European Society for Medical Oncology (ESMO; https://www.esmo.org/guidelines/guidelines-by-topic accessed on 13 December 2022) recommends the use of comprehensive cancer CGP in the following conditions:research purpose;screening for clinical trials;drug development;tumor characterization of non-squamous non-small-cell lung cancer (NSCLC), prostate cancers, ovarian cancers and cholangiocarcinoma only in cases of acceptable additional cost;tumor characterization of colon cancer as an alternative option to PCR only in cases of acceptable additional cost;tumor characterization of all cancers for which agnostic drugs are available (i.e., pembrolizumab for high tumor mutational burden, TMB).

The clinical utility of wide panels when available treatments are not approved for a specific condition is not demonstrated, and the use of off-label drugs based on genomics results should be limited to national/regional access programs [3].

In this context, the Fondazione Policlinico Universitario Agostino Gemelli IRCCS (FPG), a referral Italian research hospital, launched a CGP program (ID: FPG500, Ethical committee approval number 3837) enrolling cancer patients who were supposed to receive molecular information for treatment or prognostic definition. Profiling was carried out through a high-throughput assay named TruSight Oncology 500^TM^ (TSO500, Illumina^®^) that analyses both DNA and RNA, identifying single-nucleotide variants (SNVs), insertions/deletions (indels) and copy number variations (CNVs) in 523 genes, as well as known and unknown fusions and splicing variants in 55 genes, and provides genomic biomarkers such as microsatellite instability (MSI) and Tumor Mutational Burden (TMB), which is a measure of the number of somatic mutations present in the sequenced genome.

Given this framework, in compliance with the new EU IVD regulation IVDR 2017/746 (https://eur-lex.europa.eu/eli/reg/2017/746 accessed on 17 November 2022), an internal evaluation of the performances established by the manufacturer was performed before testing patients’ samples during clinical routine. Specifically, we reported here the identification of SNVs, indels and CNVs, comparing TSO500 with validated assays, the variability of the quality wet and sequencing metrics. All variables in both the wet bench process and bioinformatics analysis were considered.

## 2. Materials and Methods

### 2.1. Samples and Orthogonal Assay

A total of 70 DNA and 63 RNA obtained from FFPE samples of 9 different cancer types were used, as reported in Appendix A. All key cancer-specific molecular alterations which assessment is mandatory for either prognostic or therapeutic reasons were included. All samples had been previously analyzed by independent analytically validated NGS, PCR, FISH and the Sanger sequencing assay. Ethics approval for the study was obtained from the Ethics Committee Research of Catholic University of the Sacred Heart of Rome (reference ID: 3837).

DNA and RNA were extracted from 2 × 5 μm FFPE scrolls using AllPrep^®^ DNA/RNA FFPE kit (Qiagen, Hilden, Germany) following the manufacturer’s protocols. DNA and RNA concentrations were measured on a Qubit 2.0 Fluorometer (Thermo Scientific, Paisley, UK) using the Qubit dsDNA High Sensitivity and RNA High Sensitivity assay kits, respectively. Nucleic acid purity was assessed by NanoPhotometer P-Class (Implen), evaluating the ratio of the absorbance at 260 nm and 280 nm and 260 nm and 230 nm. Samples with 260/280 absorbance values between 1.6 and 1.8 for DNA and between 1.8 and 2.0 for RNA and a 260/230 absorbance ratio >2.0 were included.

The percentage of fragments >200 nucleotides in size (DV200) was assessed for RNA samples using TapeStation 4200 in association with the Agilent RNA ScreenTape kit (Agilent Technologies, Santa Clara, CA, USA). The DNA quality was determined by the Infinium HD FFPE quality control (QC) Assay Protocol (Illumina, Cambridge, UK). RNA samples with a DV200 of ≥20% and DNA samples with a Delta Cq value of ≤5 were used for downstream applications.

In addition, two commercially available reference samples were used to assess analytical sensitivity and assay processing as a run control for a total of 71 DNA and 64 RNA samples. On the DNA level, the Structural Multiplex Reference Standard HD753 (Horizon Dx) was used, which includes 10 confirmed variants (7 SNVs and 3 indels) centered at 5% VAF; moreover, it harbors RET and ROS1 fusion variants, MYC-N and MET focal amplifications. As reference material for the RNA analysis, we used Seraseq^®^ FFPE Tumor Fusion RNA v4 (SeraCare, cat# 0710-0497) harboring 16 well-known fusions and 2 exon-skipping events in EGFR (vIII) and MET (ex14 skipping).

### 2.2. Library Set-Up

Libraries were prepared using the TruSight Oncology 500 High-Throughput library preparation kit (Illumina, San Diego, CA, USA) according to the reference guide. Up to 96 ng DNA was sheared using Covaris E220 (Covaris Ltd., Woodingdean, Brighton, UK), 8 microTUBE—50 AFA Fiber Strip V2 (Covaris Ltd., Woodingdean, Brighton, UK) and Rack E220e 8 microTUBE Strip V2 (Covaris Ltd., Woodingdean, Brighton, UK). The sizes of the double-stranded DNA (dsDNA) fragments (90–250 bp) were confirmed using TapeStation 4200 (Agilent, Cheshire, UK) after shearing, with a target peak of approximately 180 bp. For fusion detection, first- and second-strand cDNA synthesis was performed starting from up to 100 ng RNA. After end repair and A-tailing, ligation of the adapters carrying the Unique Molecular Identifiers (UMIs) was performed, and fragments were amplified to add the indexes. Next, hybridization was performed overnight, followed by a streptavidin magnetic bead-based capture to enrich for the selected targets. A second 2 h hybridization and capture round was performed with the same probes; after which, the enriched fragments were amplified in a second PCR step to produce the enriched library. Purified libraries were then bead-based normalized, resulting in normalized enriched DNA- and RNA-based libraries.

Intermediate check points were performed on pre-capture and final enriched libraries via fluorometric quantification by a Qubit dsDNA High Sensitivity kit (Thermo Scientific, Paisley, UK) and analyzing the profile of each sample via capillary electrophoresis with TapeStation 4200 (Agilent, Cheshire, UK). A quantification of at least 30 ng/μL matched the size distribution fragments, and ≤250 bp is recommended for pre-capture libraries. A final enriched libraries dosage before normalization of at least 3 ng/μL is advised, with a target peak of approximately 250 bp.

Finally, the DNA- and RNA-based libraries were combined in the final library pool, containing 80% DNA and 20% RNA, denatured and diluted for instant sequencing.

Sequencing libraries of runs 1 and 2 were prepared manually, while the ones of run 3 were by an automated procedure implemented on the Hamilton Microlab STAR liquid handling system. Sequencing libraries of runs 4 and 5 were performed via both manual and automated modes.

Scripted protocols for the automated workflow were developed for all steps of TSO500 library prep, from the reverse transcription of RNA to bead-based library normalization. The number of samples per preparation was 16, and the overall samples are divided in 5 run of sequencing onto a 200-cycle format SP, S1 or S2 flowcell (based on the sample size to be processed) and sequenced via the Illumina NovaSeq 6000 platform, according to Illumina’s protocol. Runs 1 to 3 employed SP flowcells, while S1 was used for runs 4 and 5.

### 2.3. Sequencing

The TSO500 panel is designed to analyze multiple biomarkers through both DNA and RNA sequencing derived from the same patient. It encompasses 9232 regions: 7567 exons (82%), 1172 introns (12.7%), 123 Pseudogenes (1.3%) and 370 other regions (4%). In addition, it includes SNV, InDel, CNV, MSI and TMB from DNA and fusions and splice variants from RNA.

In this study, the DNA and RNA variant calling was performed using the Illumina TruSight^®^ Oncology 500 Local App v2.2 (https://support.illumina.com/downloads/trusight-oncology-500-v2-2-local-app-documentation.html accessed on 13 December 2022). The analysis reproducibility was ensured using Docker container technology [4]. After the Fastq generation, the analysis process followed the DNA and RNA phases described in Appendix A. The entire workflow included several quality controls analyses concerning the DNA and RNA metrics reported in Appendix A.

Quality metrics were set to investigate (i) the quality of the reads, (ii) the coverage on the sequenced target regions and (iii) the uncommon reads detected during the bioinformatics analysis. The quality reads analysis indicates all the quality controls related to the sequenced reads (e.g., passing filter, aligned or enriched reads). The coverage is also evaluated in several ways, such as exons mapped or depth of target. Uncommon reads detection is the analysis of chimeric reads or alternative mapping. Moreover, the Unique Molecular Identifiers (UMI) are evaluated in order to address the PCR duplicates for the both DNA and RNA analyses.

### 2.4. Bioinformatics Analysis

The DNA and RNA samples were processed using Illumina TruSigh^®^ RUO Local App v2.2. The entire analysis flow was managed by the Lianne system [5]. Lianne ensures the correct job scheduling with the Portable Batch System (PBS) on the cluster nodes, activates the Conda [6] environments with the required dependencies and manages the output data folders. The Illumina Local App was used with the Singularity container platform [7] following the steps in Appendix A.

The validation was performed by considering the ability to detect the expected molecular alterations (i.e., diagnostic accuracy) considering the variability (i.e., repeatability) of the DNA extraction, library preparation and sequencing quality metrics obtained across 5 sequencing runs of DNA. Human reference genome hg19 was used.

### 2.5. Coverage Analysis

The coverage analyses were based on a specific input coverage data format computed by Mosdepth [8], including the following fields: chromosome, region (exon) start, region (exon) end, region ID, 5X depth, 10X depth, 50X depth, 100X depth, 250X depth and 500X depth. For each region, for a given depth attribute, the number of bases covered at that depth is reported (count data). The coverage count data are then converted into frequency data (i.e., the percent coverage, *p*) divided by region length. The coverage analysis then starts from a covdata file, including the input fields and median values of *p* across samples (i.e., subjects), referred to as median percent coverage (MPC).

As a good quality principle, a run should maximize the number of depths with MPC ≥75%. Similarly, at a given depth D for each sample x, the first quartile of the exon percent coverage distribution should be ≥75% (i.e., at least 75% of the exons of x should be covered by at least 75% at depth *D*). Each run type uses different depths: 50–500X for DNA (SNVs and CNVs) and 5–50X for RNA.

The procedure of the coverage analysis is implemented in the open-source genomic data indexing and management suite VarHound [9]. In addition to the distribution of MPC per run and exon coverage across samples, VarHound returns an output table for each run type, including a per-sample diagnostic table reporting quartile values for the exon percent coverage. The median of the exon percent coverage (Q2) was used to compare coverage data to wet and sequencing metrics. A list of BED format files included blacklisted regions (i.e., exons with unreliable coverage). Keeping track of blacklisted regions is of critical importance, since they might harbor false-negative variants (i.e., possible undetected malignant traits). Additionally, a base-level analysis detects narrow coverage drops within highly covered regions (exons). As a result, a coverage drops BED file, a gene-level drops file and related Genome Browser track file are generated. These small drops might hide false-negative variant calls in apparently covered regions. The VarHound diagnostics workflow is shown in Figure 1.

### 2.6. Data Analysis

Preliminary, descriptive statistics were computed both on metrics (“wet” and sequencing) and coverage depth variables. Qualitative data were expressed as absolute frequencies, whilst quantitative variables either as the mean and standard deviation (SD) or as the median and range. The statistical units of the analysis on the wet and sequencing metrics were the samples (stratified by DNA and RNA values), whereas the coverage analysis considered the exons. The distributions of the numerical variables were graphically shown by boxplots.

Subsequently, the Kruskal–Wallis test and multiple comparisons analysis by the Mann–Whitney tests (or Brown–Mood median tests) were applied to compare wet and sequencing metric variables across the five validation runs and the coverage depth levels (50X, 100X, 250X and 500X for DNA samples and 5X, 10X and 50X for RNA ones). The false discovery rate (FDR) adjustment was applied to the *p*-values of the pairwise comparisons in order to account for type I errors. In addition, a Wilcoxon signed-rank test was applied to evaluate if pre-capture and enriched library values significantly differed by stratifying by sample type (DNA and RNA).

Next, a Kendall’s Tau rank correlation analysis between “wet” metrics and coverage (aggregated by sample), stratified by variant type (SNV, CNV and RNA) and by run (1 to 5), was also carried out. Results with *p*-values < 0.05 were statistically significant. All analyses were performed by using R software version 4.0.2 [10], custom R scripts [11] and its packages coin [12,13] and fmsb [14].

## 3. Results

Seventy-one subjects were investigated for SNVs and InDel using the TSO500 high-throughput assay and analyzed with the bioinformatic pipeline Illumina TruSigh Oncology Local App v2.2 [6]. A minimal amount recommended as input material was obtained for all samples, except for one RNA sample that had a lower input. Additionally, three RNA libraries were excluded from the final sequencing, because they did not reach the minimal concentration advised.

### 3.1. Variant Results

The expected DNA molecular alterations were derived from the Gemelli data warehouse and were detected using different methodologies. Further details about the expected and detected alterations are available in Appendix A. All expected small DNA alterations (SNV, InDel) were found by the Illumina TSO500 process, excluding the nonsense variant in the TSO500_D068 sample. In addition, the Variant Allele Frequency (VAF) was evaluated for the samples performed previously with the NGS methodology, and only the nonsense variant in the TSO500_D042 BRCA2 sample showed a discrepancy between TSO500 and the expected result. Patients with suspected germinal variants following VAF evaluation were redirected to genetic counseling. Fourteen out of twenty CNV alterations were not detected; the missed detections involved both germline and tissue CNV BRCA1/2 alterations. All variants included in the Horizon Positive control were detected. In addition, data regarding RNA, MTB and MSI were not reported, since no comparative assay was available for this study.

### 3.2. Bioinformatics Analysis

The analyzed samples were 71 on DNA and 64 on RNA, including the two control templates. Five RNA samples were discarded from the analysis: four samples failed the quality checks of the enriched libraries, and they were not sequenced because they were not compliant with the wet metrics (4), while one sample returned missing values after sequencing.

Table 1 and Table 2 show wet quality metrics descriptive statistics for both DNA and RNA. Concerning DNA, significant differences across runs were observed for all metrics but DNA abundance. Notably, the median value of the enriched libraries significantly differed between the DNA and RNA samples (18.5 vs. 6.51, *p* < 0.001), whereas the median pre-capture libraries values were different in a suggestive way (48.7 vs. 51, *p* = 0.064). In addition, in both the DNA and RNA samples, the pre-capture and enriched library metrics showed a significant decrease in median values. Concerning the DNA sequencing metrics, the mean of the median target coverage was 648.1 ± 324.13 units, and the median insert size was compliant in relation to its threshold (≥70, 71/71) and by comparing mean and median values equal to 110.3 (±16.45) and 113.

In addition, exon 100X (95.76 ± 8.56), Target 100X (95.09 ± 9.03) and Target 250X (82.49 ± 24.88) suggested a reliable sequencing process for both the coding (exon) and target regions (Appendix A).

Finally, the coverage MAD (median absolute deviation; i.e., the median normalized deviation across all regions used for CNV calling) provided acceptable evidence both in relation to the compliance threshold (≤0.21, 61/71 = 86%) and mean and median values equal to 0.164 (±0.047) and 0.161, respectively.

The correlation results between coverage and wet metrics were reported respectively for DNA (Table 3) and RNA (Table 4).

Concerning the RNA samples, a small fraction of them was affected by coverage depletion at 5–10X (3/59 and 4/59 samples, respectively; see Appendix A) and a higher fraction at 50X (13/59 samples). Moreover, the RNA sample coverage is generally limited by the smaller quantity of the initial nucleic acid concentration strong correlation between the A260 and 280 or A260 and 230 ratios and RNA coverage at 5–50X (Table 4).

Figure 2 and Figure 3 show the boxplots of the sequencing quality metrics by run for DNA and RNA, respectively (original data from Appendix A). For each metric, the panels show significant differences between runs (Wilcoxon rank-sum test *p*-value < 0.05). Despite these differences, all runs showed metrics above the platform guidelines [6].

In addition, Figure 4 shows the radar chart of the relevant sequencing coverage and mapping quality metrics. Both the DNA and RNA samples show low levels of chimeric reads and a high depth of coverage.

The correlation patterns between the wet metrics and coverage for both DNA (Table 3 and Figure 5) and RNA (Table 4 and Figure 6) samples were evaluated by Kendall’s tau correlations. Both tables and heatmaps showed high-coverage metrics correlations measured across the regions designed for CNV and SNV detection. On the other hand, RNA metrics showed a consistent coverage correlation from 5X to 50X.

## 4. Discussion

The goal of this study was to evaluate the performance of the Illumina TSO500 HT platform on a set of critical SNVs and CNVs, representing key prognostic biomarkers and targets for personalized therapy. In addition, we studied the correlation between the Illumina wet [11], sequencing [5] and coverage metrics [9], providing an open-source set of bioinformatics tools for the TSO500 assessment.

Our results indicate that TSO500 is a reliable test with a robust workflow both for the wet and for the computational steps in the SNV and InDel analyses. All expected alteration were detected, and a good coverage performance was provided: high values of the median coverage, coverage MAD, median insert size, exon 100X and target at 100X and 250X probed the reliability of the bioinformatics pipeline, useful to call SNV and indel variants, for both the exon and target regions [1,2]. Accounting for this, no false negatives were detected in the small variants calling. These high performances were expected and in line with the recent literature reports [2,15,16].

On the other hand, the CNV analysis showed major limitations, identifying only six out of twenty alterations, with BRCA1 not detected at the exon resolution (samples D036 and D037). The Illumina Local App algorithm is not designed for an exon resolution analysis, thus detecting only complete gene deletions. Consequently, we recommend caution when using this panel for the detection of clinically relevant CNVs. Although the recent literature considered this panel generally safe for CNV detection [2,15,16], we found coverage-related issues that could be worthy of attention.

Regarding the coverage analysis, two broad profile types emerged: a high-coverage profile, characterized by a flat exon plateau with gradual decay beyond the exon–intron boundaries and a low-coverage profile, with an irregular or discontinuous plateau and coverage breakoffs starting before the exon boundaries, generating large portions of signal depletion (nonuniform coverage profile). Our exon MPC measurements showed that, at high depths (250–500X), coverage-depleted regions could involve entire exons, and large parts of a gene (Appendix A, depth 250–500X), hampering SNV and CNV calling, even if the global coverage metrics are above the guideline values (Appendix A). As previously reported [17,18], low coverage and coverage nonuniformity could deeply affect the whole-exome sequencing performances, especially for CNV detection, often based on the detection of continuous depleted or enriched adjacent exons [17]. This issue could be further intensified in long exons, where nonuniformity might cause a failure in calculating the background coverage [17]. In addition, it is widely recognized how a critical contribution to incomplete sequencing is a reduced coverage in regions showing a drop in high-quality mapped reads, potentially affecting the variant calling performances [18].

However, low-coverage issues are strongly restrained at medium-low depths (Appendix A, depth 50–100X), making this technology safe for both SNV/indel and CNV calling, unless possible low-coverage regions are monitored at the gene and exon level for each sample through a dedicated bioinformatic protocol (Figure 1). By profiling the sample-level median exon coverage, we showed how depleted regions are rare and affect only a tiny fraction of the DNA samples at 50X and 100X (0/71 and 1/71 samples, respectively, for both SNV/indels and CNV coverage; see Appendix A).

Analogous considerations can be done for RNA samples, a small fraction of which was affected by coverage depletion at 5–10X (3/59 and 4/59 samples, respectively; see Appendix A) and a higher fraction at 50X (13/59 samples). Therefore, at greater depths, the TSO500 RNA samples might suffer from low coverage, possibly affecting RNA fusion detection. RNA sample coverage is generally limited by an inferior quantity of the initial nucleic acid concentration compared to the DNA samples. This was highlighted by the strong correlation between the A260 and 280 or A260 and 230 ratios and RNA coverage at 5–50X (Table 4 and Figure 6).

Finally, regarding the repeatability, even if these key metrics provided evidence of variability across the run, the high medians, the compliant values in relation to the guideline thresholds and the low intra-run variability validated the process.

## 5. Conclusions

This study showed a 100% agreement between TSO500 and the standard approaches in detecting key cancer-specific molecular alterations, meaning it is a reliable assay for clinical practice. The current limits in CNV variants and intragenic analyses must be addressed before using the assay in ovarian, pancreatic and prostate cancer, which require accurate assessments of the BRCA 1/2 genes. Finally, we highlighted the importance of accurately monitoring coverage-depleted and nonuniform coverage profiles, since they could negatively affect the SNV and CNV detection ability of this panel.

## Figures and Tables

**Figure 1 cancers-14-06152-f001:**
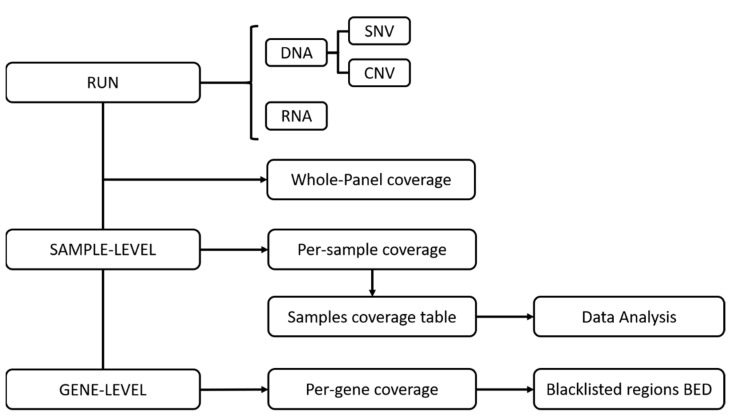
Coverage analysis workflow.

**Figure 2 cancers-14-06152-f002:**
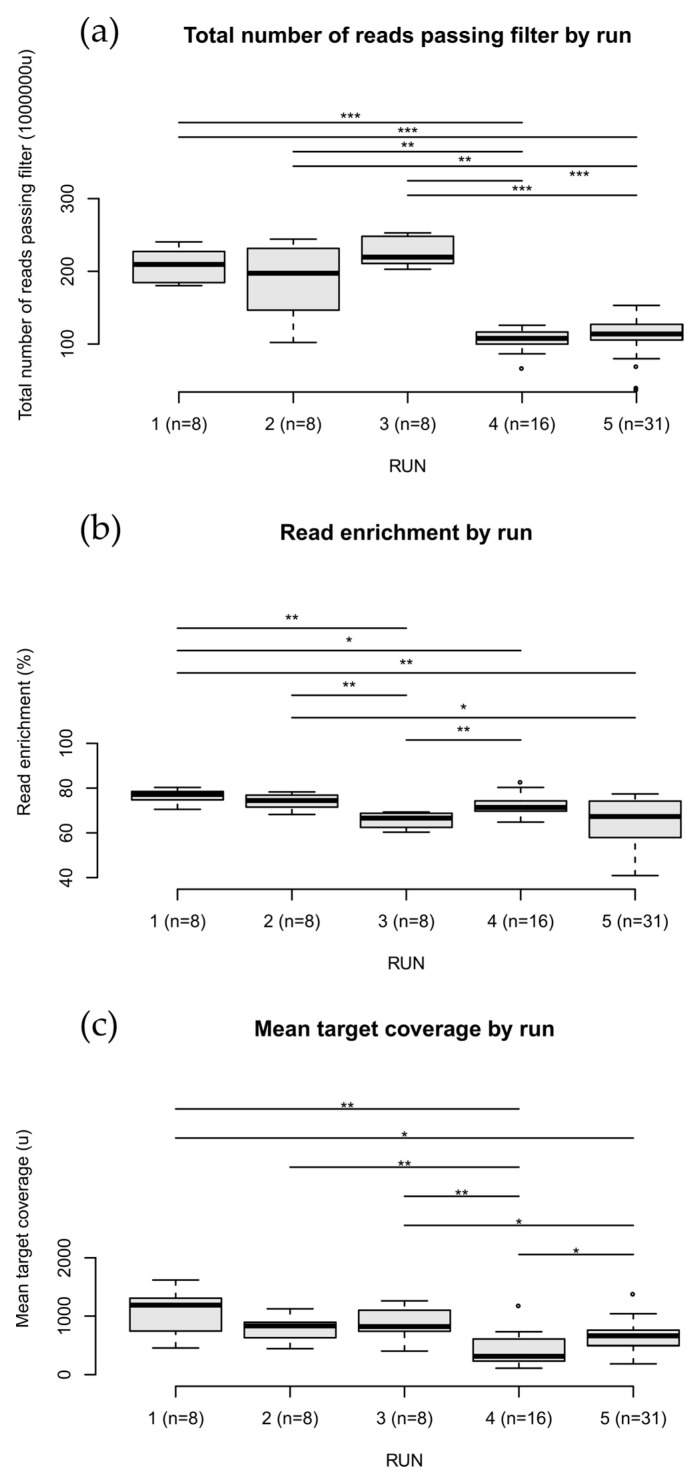
DNA sequencing quality metrics over 71 samples. Each boxplot shows the metrics distribution for the five runs. Pairwise comparisons between runs were done through a two-sided Wilcoxon rank-sum test (*p*-value: *** < 0.001, ** < 0.01 and * < 0.05). (**a**) Total number of reads passing (quality score > 30). (**b**) Read enrichment as the percentage of reads aligned on target over the total aligned ones. (**c**) Mean read coverage on the panel target probes.

**Figure 3 cancers-14-06152-f003:**
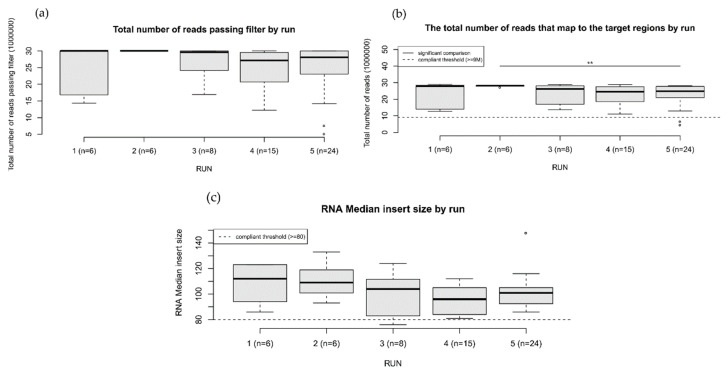
RNA sequencing quality metrics over 59 samples. Each boxplot shows the metrics distribution for the five runs. Pairwise comparisons between runs were done through a two-sided Wilcoxon rank-sum test (*p*-value: ** < 0.01). (**a**) Total number of reads passing (quality score > 30). (**b**) Total number of reads mapping to the target regions. (**c**) Mean read length (bp) by run.

**Figure 4 cancers-14-06152-f004:**
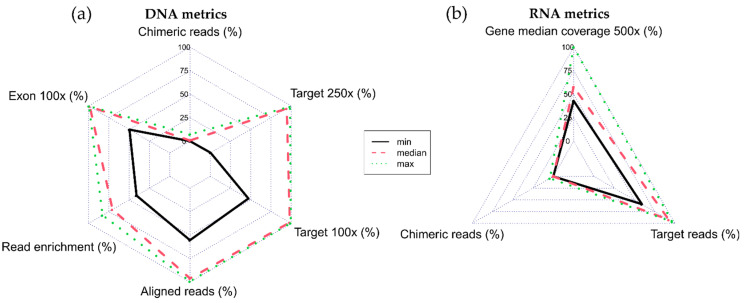
Radar plot reporting sequencing coverage and mapping quality metrics for 71 DNA (**a**) and 59 RNA (**b**) samples.

**Figure 5 cancers-14-06152-f005:**
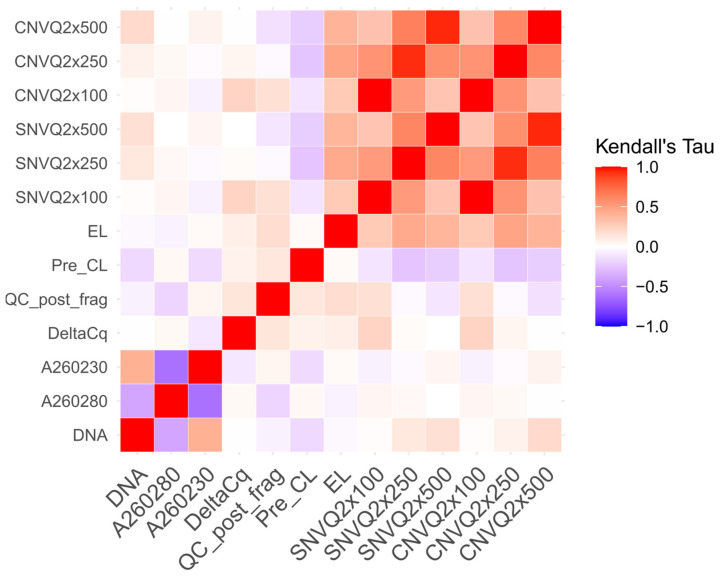
Kendall’s tau correlation coefficients between the DNA wet and sequencing coverage metrics.

**Figure 6 cancers-14-06152-f006:**
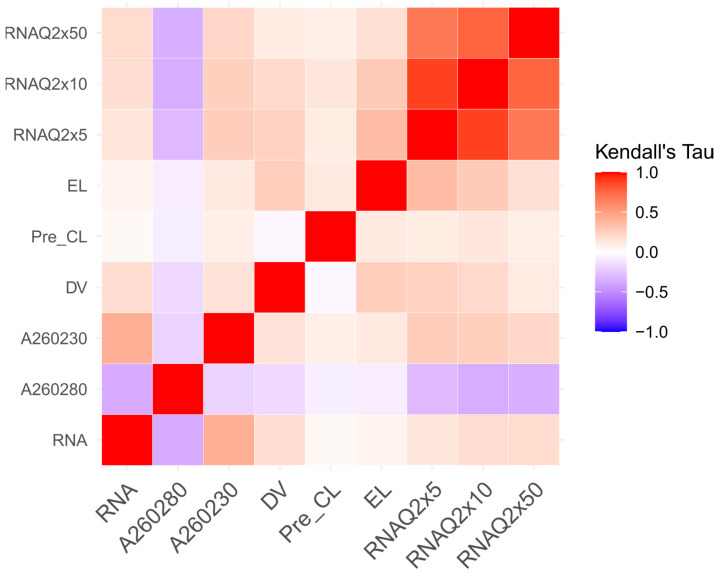
Kendall’s tau correlation coefficients between the RNA wet and sequencing coverage metrics.

**Table 1 cancers-14-06152-t001:** DNA wet quality metrics. Quality metrics values, for each run, are reported as mean ± standard deviation and median [range] of the 71 DNA samples. A Kruskal–Wallis test is used to evaluate the null hypothesis of the equality of the medians among the runs. Median values are reported in bold. *p*-values (*p*) below 0.05 are considered significant.

	Overall	Run 1	Run 2	Run 3	Run 4	Run 5	*p*
DNA (ng/ul) (QT) (CV if >3.5)	34.58 ± 52.50 **17.1** [3.11; 298]	74.14 ± 86.11 **24.45** [9.40; 242]	16.448 ± 10.52 **14.25** [5.4; 36.7]	25.66 ± 27.13 **17.75** [4.4; 88.7]	44.792 ± 74.04 **14.85** [3.11; 298]	26.07 ± 33.28 **17.1** [3.4; 184]	0.598
A 260/280 (u) (QL) (CV if >2)	2.02 ± 0.219 **1.97** [1.50; 3.06]	2.058 ± 0.093 **2.045** [1.93; 2.21]	2.186 ± 0.145 **2.195** [1.97; 2.36]	2.107 ± 0.393 **1.965** [1.9; 3.06]	1.921 ± 0.141 **1.9** [1.5; 2.12]	1.997 ± 0.209 **1.95** [1.58; 2.6]	**0.004**
A 260/230 (u) (QL) (CV if >2)	0.884 ± 0.684 **0.67** [0.1; 2.28]	0.59 ± 0.448 **0.38** [0.23; 1.49]	0.303 ± 0.162 **0.25** [0.18; 0.67]	1.439 ± 0.909 **2** [0.1; 2.28]	1.183 ± 0.784 **1.125** [0.26; 2.2]	0.814 ± 0.539 **0.75** [0.14; 2.12]	**0.005**
Delta Cq (u) (QL) (CV if <5)	0.192 ± 1.541 **0.3** [−4.7; 3.6]	0.838 ± 0.851 **0.45** [0.1; 2.5]	0.437 ± 0.722 **0.15** [−0.4; 1.7]	−0.185 ± 1.369 **−0.5** [−1.6; 2]	−1.181 ± 1.818 **−0.7** [−4.7; 1.4]	0.769 ± 1.282 **0.8** [−2.61; 3.6]	**0.001**
Quality Control post Fragmentation (bp) (QT) *	235.1 ± 35.73 **232** [173; 315]	197.9 ± 17.59 **207.5** [173; 216]	197.1 ± 13.23 **195.5** [181; 212]	220.8 ± 25.49 **219** [187; 265]	226.8 ± 28.62 **228.5** [174; 272]	262.4 ± 26.21 **261** [203; 315]	**<0.001**
Pre-capture libraries metric (ng/ul) (QL) (CV if >20)	47.43 ± 7.298 **48.7** [26.4; 60]	46.64 ± 3.613 **47** [41.6; 52]	49.85 ± 1.790 **49.65** [47.7; 53]	30.7 ± 3.001 **30.45** [26.4; 35.6]	49.71 ± 3.373 **49.25** [44.4; 57]	50.16 ± 5.089 **50** [39.9; 60]	**<0.001**
Enriched libraries metric (ng/ul) (QL) (CV if >3)	16.7 ± 7.959 **18.5** [1.53; 31.9]	12.161 ± 4.368 **13.95** [5.58; 16.4]	17.49 ± 8.652 **20.6** [3.4; 25.8]	19.06 ± 1.694 **18.6** [17; 21.7]	10.125 ± 5.233 **8.96** [3.34; 21.6]	20.45 ± 8.173 **23** [1.53; 31.9]	**<0.001**

Keys: CV: compliant value; QT: variable of quantification; QL: variable of qualification; u: units; QC: Quality Control; NA: Not Available (i.e., QC not performed); NE: Not Expected; u: units; KW: Kruskal–Wallis test; * Compliant with Illumina guidelines if included between 150 and 300.

**Table 2 cancers-14-06152-t002:** RNA wet quality metrics. Quality metrics values, for each run, are reported as mean ± standard deviation and median [range] of the 59 RNA samples. A Kruskal–Wallis test is used to evaluate the null hypothesis of the equality of the medians among the runs. Median values are reported in bold. *p*-values (*p*) below 0.05 are considered significant.

	Overall	Run 1	Run 2	Run 3	Run 4	Run 5	*p*
RNA (ng/ul) (QT) (CV if >10.5)	78.3 ± 63.40 **66** [12.3; 312]	70.33 ± 41.27 **70.95** [23.7; 120]	131.4 ± 105.13 **89.7** [30.1; 312]	46.1 ± 52.67 **30.65** [12.3; 170]	104.74 ± 58.11 **86** [36; 235.8]	61.23 ± 50.90 **45.5** [13.1; 200.1]	**0.005**
RNA A260/280 (QL) (CV if >2)	1.942 ± 0.101 **1.96** [1.6; 2.2]	2 ± 0.109 **2** [1.9; 2.2]	1.917 ± 0.098 **1.95** [1.8; 2]	1.976 ± 0.089 **2** [1.76; 2.05]	1.871 ± 0.066 **1.9** [1.7; 1.97]	1.968 ± 0.102 **1.98** [1.6; 2.2]	**<0.001**
RNA A260/230 (QL) (CV if >2)	1.123 ± 0.635 **1.36** [0.03; 1.95]	1.105 ± 0.563 **1.21** [0.09; 1.6]	1.113 ± 0.609 **1.1** [0.17; 1.75]	0.89 ± 0.817 **0.87** [0.03; 1.95]	1.66 ± 0.314 **1.7** [0.6; 1.9]	0.871 ± 0.575 **0.79** [0.16; 1.94]	**0.001**
DV200 (%) (QL) (CV if >20)	59.2 ± 16.71 **63.7** [2.6; 86.9]	59.33 ± 15.49 **64.7** [36.3; 73.2]	60.32 ± 17.52 **56.6** [41.7; 83.5]	61.29 ± 18.73 **68.1** [33.2; 82.5]	65.25 ± 10.79 **67.2** [42.6; 77.8]	54.4 ± 18.92 **55.95** [2.6; 86.9]	0.455
Pre-capture libraries metric (ng/ul) (QL) (CV if >20)	48.84 ± 8.381 **51** [25.7; 60]	48.12 ± 2.778 **47.4** [45.4; 53]	45.93 ± 2.809 **45.3** [43; 51]	31.15 ± 3.787 **30.85** [25.7; 36.4]	52.05 ± 2.775 **53** [46.1; 55]	53.63 ± 4.780 **54.5** [39.7; 60]	**<0.001**
Enriched libraries metric (ng/ul) (QL) (CV if >3)	7.033 ± 4.702 **6.51** [0.80; 19.6]	5.145 ± 3.821 **5.755** [0.8; 9.03]	9.685 ± 6.081 **7.45** [3.8; 18.6]	6.893 ± 6.023 **5.615** [1.07; 17]	6.981 ± 5.636 **5.25** [1.12; 19.6]	6.920 ± 3.372 **6.925** [2.1; 13.8]	0.635

QT: variable of quantification; QL: variable of qualification; u: units; QC: Quality Control; NA: Not Available (i.e., QC not performed); NE: Not Expected; u: units; KW: Kruskal–Wallis test.

**Table 3 cancers-14-06152-t003:** Correlation analysis between the median coverage and wet metrics for the DNA samples. Correlations were evaluated through a tau test, evaluating the Kendal’s rank correlation (tau) significance between wet metrics and coverage at three different depths (100X, 250X and 500X) for SNV and CNV. Each cell reports: Kendall’s tau coefficient, *p*-value and tau 95% confidence interval. Significant correlations (*p*-value < 0.05) are highlighted in bold.

	SNV 100X	SNV 250X	SNV 500X	CNV 100X	CNV 250X	CNV 500X
DNA	0.013 0.891 −0.222, 0.249	0.126 0.181 −0.076, 0.328	0.170 0.060 −0.009, 0.350	0.013 0.891 −0.222, 0.249	0.067 0.474 −0.138, 0.274	0.196 **0.029** 0.016, 0.375
A 260/280	0.047 0.634 −0.158, 0.252	0.035 0.710 −0.145, 0.217	0 1 −0.180, 0.180	0.047 0.634 −0.158, 0.253	0.033 0.727 −0.149, 0.216	−0.010 0.913 −0.190, 0.170
A 260/230	−0.058 0.549 −0.278, 0.161	−0.027 0.774 −0.195, 0.141	0.051 0.578 −0.128, 0.229	−0.058 0.550 −0.278, 0.161	−0.019 0.836 −0.188, 0.149	0.062 0.489 −0.113, 0.238
Delta Cq	0.230 **0.019** 0.011, 0.449	0.022 0.812 −0.188, 0.234	−0.001 0.991 −0.191, 0.189	0.230 **0.019** 0.012, 0.449	0.048 0.615 −0.166, 0.262	−0.008 0.926 −0.196, 0.179
Quality Control post Fragmentation	0.170 0.081 −0.053, 0.393	−0.031 0.737 −0.224, 0.161	−0.114 0.201 −0.287, 0.058	0.170 0.081 −0.053, 0.393	−0.028 0.762 −0.224, 0.167	−0.136 0.130 −0.311, 0.037
Pre-capture libraries metric	−0.120 0.221 −0.323, 0.083	−0.257 **0.007** −0.420, −0.094	−0.216 **0.018** −0.374, −0.058	−0.120 0.222 −0.323, 0.083	−0.250 **0.008** −0.419, −0.080	−0.216 **0.017** −0.373, −0.059
Enriched libraries metric	0.283 **0.003** 0.032, 0.534	0.450 **<0.001** 0.280, 0.620	0.391 **<0.001** 0.245, 0.536	0.283 **0.003** 0.032, 0.534	0.474 **<0.001** 0.314, 0.633	0.392 **<0.001** 0.238, 0.545

**Table 4 cancers-14-06152-t004:** Correlation analysis between the median coverage and wet metrics for the RNA samples. Correlations were evaluated through a tau test, evaluating the Kendal’s rank correlation coefficient (tau) significance between the wet metrics and coverage at three different depths (5X, 10X and 50X). Each cell reports: Kendall’s tau coefficient, *p*-value and tau 95% confidence interval. Significant correlations (*p*-value < 0.05) are highlighted in bold.

	5X	10X	50X
RNA	0.141 0.144 −0.047; 0.329	0.175 0.062 −0.207; 0.559	0.188 **0.036** 0.039; 0.336
RNA A260/280	−0.306 **0.002** −0.473; −0.138	−0.361 **<0.001** −0.531; −0.192	−0.344 **<0.001** −0.507; −0.181
RNA A260/230	0.265 **0.006** 0.087; 0.445	0.252 **0.008** 0.073; 0.430	0.218 **0.016** 0.050; 0.386
DV200	0.249 **0.010** 0.082; 0.415	0.202 **0.032** 0.039; 0.366	0.109 0.226 −0.058; 0.275
Pre-capture libraries metric	0.105 0.284 −0.080; 0.290	0.132 0.165 −0.041; 0.306	0.093 0.303 −0.084; 0.272
Enriched libraries metric	0.356 **<0.001** 0.184; 0.528	0.279 **0.003** 0.102; 0.456	0.172 0.056 −0.015; 0.358

## Data Availability

The data presented in this study are available on request from the corresponding author. The data are not publicly available due to their containing information that could compromise the privacy of research participants.

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
