# Peer review of "A Computational Framework for Comprehensive Genomic Profiling in Solid Cancers: The Analytical Performance of a High-Throughput Assay for Small and Copy Number Variants"

_cancers, 2022, doi:10.3390/cancers14246152_

Round 1

Reviewer 1 Report

The authors report on the analytical performance of the CGP TSO500 assay for routine screening of patients  through an internal validation on 71 DNA and 64 RNA samples in 5 NGS runs. The focus is on the computational analysis of panel assessment and data processing mainly by comparing the quality of the wet lab variables with those of the sequencing metrics. Though the study is novel it lacks critical data that is key to the main goal of a clinical validation study.

Major comment:

An analytical performance validation of an assay for its integration in a clinical setting requires the analysis of several performance characteristics including precision, sensitivity, specificity, accuracy and limit-of-detection.  In this manuscript, only the sensitivity/specificity/accuracy parameters has been analyzed for SNV and indel detection. However, most analyzed variants were in a few genes and the variants that were compared with an orthogonal NGS method were all in BRCA1/2 or few other BRCA-ness genes. In addition, based on the VAF these might even not be somatic variants. As the TSO500 assay is known not to be able to detect intragenic CNVs it is not surprising that such CNV's in BRCA1/2 were missed. Whole gene amplifications, which can be detected, were not tested here. Finally, gene fusions, TMB and MSI values, that also make part of the assay, were not discussed as they could not be compared with data from an orthogonal method. Therefore, the paper cannot be called an internal validation with analytical performance testing.

Other comments:

It is generally known that DNA and RNA of low quality is likely to result in poor data. 

It is hard to compare data of 5 NovaSeq runs, which include a different number of samples each and use other flowcells. Moreover, it is unclear which flowcell was used for each run. It will have an impact on the median coverage per sample.

Quality threshold are given for inclusion of DNA and RNA samples (OD, DV200 and dCt) but based on Table 1 and 2 these thresholds are not always met for the OD ratio's. Why are these samples still included?

Are UMI's only taken into account for RNA analysis (pg4)? Why not for DNA analysis?

Values of Y axes of Figure 2 should be adapted to the data, not to the horizontal significance lines.

Some unclear communications such as: ... were different in a suggestive way  ... ; the library-related process ... ; 

Figure 4a and b: boxplot of run 2 is reduced to a line. Is that correct?

What is the evidence that the hybrid library prep would have an effect on the results? Too few runs are performed for this assumption.

Correlation of OD with RNA coverage, causing poor coverage profiles at higher depth. Is that correct?

MSI and TMB are biomarkers, not signatures. Mutation signatures mean something completely different.

Several references refer to protocols and manuals and thus should not be in the reference list.

In the discussion, only 2 references are provided. However, the aim of the discussion is to compare the data of the manuscript with that of existing literature, which is completely missing here. Notably, reference 2 cannot be used to affirm the finding that coverage-depleted regions in exons will hamper SNV and CNV calling. To my knowledge, this validation paper does not mention that.

Reviewer 2 Report

I have reviewed the manuscript “A Computational Framework for Comprehensive Genomic Profiling in Solid Cancers: The Analytical Performance of a High-Throughput Assay” by Giaco et al submitted for publication in cancers (MDPI).

Summary

In the study, the authors aimed to evaluate the performance of the TruSight Oncology 500 Assay provided by Illumina in 71 DNA and 64 RNA FFPE samples in a clinical context. They investigated several criteria for assessment of quality of the library preparation and sequencing results for both DNA and RNA. They presented several Bioinformatics open source solutions for coverage analysis and reported results for small variants, indels and copy numbers. The authors concluded that TruSight Oncology 500 is a reliable assay for clinical practice.

Strengths

The overall study design and methods are properly designed. The study provided a general framework for panel validation in a clinical context. The correlations between wet and sequencing metrics are explored.

Weaknesses and comments

1. The authors evaluated the performance of the TruSight Oncology 500 assay only by its ability to detect the expected mutations and copy number alterations. This is clearly not enough. Assay performance must be evaluated with accuracy, F1 score, positive predictive value (PPV), and or percentage of positive accordance (PPA) in order to take into account the true positives (TP), true negatives (TN), false positives (FP) and false negatives (FN).

2. TruSight Oncology 500 now reports HRD (Homologous Recombination Deficiency). This is not discussed in this paper, why ?

3. In some of the references, there are mismatches.

For example ref for Lianne system [15] , conda [16] and Docker [14] are wrong.

I suggest authors should recheck all their references using a reference manager to incorporate them without any errors.

4. In Table 1, please add the meaning for CV

5. Supplementary Table 2, the copy number alterations are not well described. I guess, CNV+ means gain/amplification, gCNV means germline copy number, and tCNV means somatic. In this case, what is the rationale to report gains in BRCA genes for Ovarian cancers in a clinical context ?

The authors should explain the meaning of the abbreviation or detail the copy number alterations.

6. In the section 2.4. Bioinformatics Analysis, the authors should specify the human reference genome version and the transcriptome database used for the analysis of DNA and RNA.

7. Supplementary Table S3. Could you explain the contamination score ? Is it cross samples contamination or DNA contamination from other species (bacteria, mycoplasma etc …) ?

8. In the section 3.1. The authors said : “In addition, data regarding RNA, MTB and MSI were reported without any comparative assays” Where are the results of Fusions (RNA), MTB and MSI ?

9. In the section 3.2. “Moreover, RNA sample coverage is generally limited by thesmaller quantity of initial nucleic acid concentration strong correlation between A260/280 or A260/230 ratios and RNA coverage at 5-50x (Table 2)”. I think, it's Table 2 not Table 4.

10. It's well known that the detection of CNV with panel sequencing is a challenge. This is particularly true for copy number deletion in bad FFPE samples. The authors should try another software to detect these alterations.

11. The Run 4 (Hybrid) showed the worst metrics in terms of aligned reads (%), insert size and coverage. It will be interesting to explain that. Is it due to bad FFPE samples or maybe the library preparation process ?

Reviewer 3 Report

The manuscript submitted by Luciano Giacò and colleagues performed validating the NGS panel in the 71 DNA and 64 RNA samples derived from various cancer FFPE tissues for clinical practice using TruSight Oncology 500 Assay (TSO500). The study showed that TSO500 has 100% agreement between TSO500 and standard approaches in detecting key cancer-specific molecular alterations, but has limited CNV variants and intragenic analysis for the assessment of BRCA 1/ 2 genes accurately. The manuscript is concisely written with clear objectives and the outcome of the study. Overall comprehensive performance validating and data analysis including wet-quality metrics were performed well. The authors particularly showed that sample purity is an essential factor for the data quality in the RNA samples. However, this study did not show the performance of fusion gene detection using RNA in the TSO500 panel.

1.       What does the MAD stand for?

2.       The fourth paragraph on page 8, “Concerning the RNA samples, a small fraction of them were affected by coverage depletion at 5-10x (3/59 and 4/59 samples, respectively; see Table S3), and a higher fraction at 50x (13/59 samples).” There is no this information in Table S3.

3.       It would be better to discuss more on the correlation between sample purity and poor coverage in RNA samples.

Round 2

Reviewer 1 Report

This paper provides interesting data  on the statistical analysis of DNA and RNA quality versus QC parameters of the samples analyzed with the TSO500 assay. Validation data on the SNV and CNV detection is also provided. Gene fusion detection, and TMB and MSI analysis, which also make part of this Comprehensive assay, was not performed.

Not having an orthogonal assay to assess the limit of detection is not an excuse not to perform these assays. It is very important to determine the % VAF threshold, which is not investigated here.

The statement 'These limitations have been extensively explained in the revised version of our Discussion section (lines 494-549 of the revised manuscript)' is strange as the revised manuscript only has 461 lines.

Figure 4a and b: boxplot of run 2 is reduced to one line because the 6 samples all had the same values. Why would that be the case in this run and not in the other runs? There should be an explanation for this uncommon finding.

Reviewer 2 Report

I am satisfied with the author’s responses to my questions raised in my initial review.